# Assessment of the Biodegradability and Compostability of Finished Leathers: Analysis Using Spectroscopy and Thermal Methods

**DOI:** 10.3390/polym16131908

**Published:** 2024-07-03

**Authors:** Alberto Vico, Maria I. Maestre-Lopez, Francisca Arán-Ais, Elena Orgilés-Calpena, Marcelo Bertazzo, Frutos C. Marhuenda-Egea

**Affiliations:** 1Footwear Technological Institute (INESCOP), C/Alemania 102—Polígono Campo Alto, 03600 Elda, Spaineorgilles@inescop.es (E.O.-C.); mbertazzo@inescop.es (M.B.); 2Department of Biochemistry and Molecular Biology and Agricultural Chemistry and Edafology, University of Alicante, Carretera San Vicente del Raspeig s/n, 03690 Alicante, Spain

**Keywords:** biodegradation, composting, leather finishing, bio-based materials, FT-IR, thermogravimetry, CP-MAS

## Abstract

In this study, the biodegradation properties of leather treated with various finishing chemicals were evaluated in order to enhance the sustainability of leather processing. We applied advanced analytical techniques, including FT-IR, thermogravimetric analysis (TGA), and solid-state NMR spectroscopy. Leather samples treated with different polymers, resins, bio-based materials, and traditional finishing agents were subjected to a composting process under controlled conditions to measure their biodegradability. The findings revealed that bio-based polyurethane finishes and acrylic wax exhibited biodegradability, while traditional chemical finishes like isocyanate and nitrocellulose lacquer showed moderate biodegradation levels. The results indicated significant differences in the biodegradation rates and the impact on plant germination and growth. Some materials, such as black pigment, nitrocellulose lacquer and wax, were beneficial for plant growth, while others, such as polyurethane materials, had adverse effects. These results support the use of eco-friendly finishes to reduce the environmental footprint of leather production. Overall, this study underscores the importance of selecting sustainable finishing chemicals to promote eco-friendly leather-manufacturing practices.

## 1. Introduction

In Europe, the leather-and-related-products sector comprises approximately 36,000 companies, employing around 435,000 people and generating a turnover of EUR 48,000 million as of 2021. The primary applications of EU leather production include footwear (41%), leather goods (19%), furniture (17%), and in the automotive industry (13%) [1]. The EU Strategy for Sustainable and Circular Textiles [2] outlines pathways to support the textile, leather, and footwear sectors toward transitioning to a modern, greener, more competitive production model. Starting in 2025, European companies, especially those operating in footwear, leather, and textile production, will have to comply with European directives on waste and the circular economy, which include managing the organic waste generated in their production processes rather than sending it to landfill.

The leather industry plays a role in recovering part of the waste generated in the meat industry; leather production is the most common option for utilizing collagen-based biomaterials, with over 20 million cattle hides processed annually [3]. This makes leather a sustainably sourced raw material. Rawhides or skins undergo various mechanical and chemical processes, including hair removal, trimming, soaking, liming, fat liquoring, and tanning, to produce durable and sustainable finished leather products. However, the use of chromium(III) salts in tanning processes, employed by more than 85% of the global leather industry, raises environmental and health concerns due to their potential oxidation to carcinogenic Cr(VI) under certain conditions [4,5,6]. To address these issues, alternative tanning processes using vegetable tannins [4], zeolites [5], and other metal-free agents [6] have been developed to reduce the environmental impact and improve the biodegradability. Additionally, the finishing processes for leather involve the application of chemicals, primarily polymeric agents such as polyacrylates and polyurethanes, to increase their aesthetic and functional properties. However, the shift toward water-based finishes to meet environmental requirements has led to the use of toxic crosslinking agents [7,8], necessitating the exploration of biodegradable and less toxic alternatives. The search for environmentally friendly and human-health-friendly products has paved the way for the development of bio-based finishing products, although their biodegradability is not always as good as one might expect.

Biodegradation, the breakdown of organic compounds by microorganisms and their transformation into CO_2_ and water, along with other by-products of the reaction, is influenced by the substrate type, environmental conditions, and microbial populations. Composting offers a sustainable solution for managing organic waste, reducing landfill waste, enriching soil, and mitigating greenhouse gas emissions [9]. However, the influence of the finishing ingredients on leather biodegradability remains unknown and requires further research. Often, the physical or chemical degradation of a polymer may resemble the changes produced due to biodegradation but may not actually be caused by the action of microorganisms during composting, highlighting the need to employ more sophisticated and specific evaluation methods in order to quantitatively determine the biodegradability of polymeric materials in compostable environments.

Laboratory-scale composting can play a crucial role as a preliminary step in the research and development of new biodegradable materials by allowing the development of a controlled process in terms of the humidity, temperature, and air flow. These biodegradation conditions are more influential than others. Currently, there are no scientific studies available regarding the actual compostability of waste from industries working with tanned hides and skins. However, there are studies on waste obtained from other stages of the tanning cycle, such as the composting of waste from the unhairing and liming processes, and studies have also been conducted on solid hair waste as a by-product of the leather-manufacturing process [10]. Although studies are still in the early stages, it is expected that composting will become one of the most important end-of-life options for managing textile waste in general, and leather in particular, in the near future.

Laboratory-scale composting studies based on international standards, such as ISO 14855-1:2012 [1], designed for plastic polymers, provide insights into leather biodegradability under controlled conditions. The standard mentioned above was designed with the intention of simulating typical aerobic composting conditions by exposing the test material to an inoculum derived from the organic fraction of solid mixed municipal waste. Composting takes place in an environment in which the temperature, aeration, and humidity are accurately monitored and controlled, while the percentage of evolved CO_2_ is quantified by employing direct measurements, such as infrared spectroscopy. Additionally, it is crucial that the quality and maturity of the compost obtained are assessed in order to determine its suitability for agricultural use and to identify any potential toxic compounds that could restrict its application [11,12,13].

In this study, we assessed the biodegradability of leather-finishing products using techniques such as FT-IR spectroscopy, TGA, and NMR spectroscopy. Samples were selected based on their significance in the footwear and leather industry, as well as their chemical composition. The products and leathers treated with various finishing agents were subjected to controlled composting on the laboratory scale. The aim was to identify sustainable treatments that enhance the biodegradability of leather production and reduce its environmental impact.

## 2. Materials and Methods

### 2.1. Composting Procedure

The composting experiment was conducted at the Laboratory of Biotechnology of the Footwear Technology Centre INESCOP (Elda, Alicante), following the controlled laboratory composting conditions according to the ISO14855:2013-1 standard [14]. The compost system provided by ECHO Instruments (Slovenske Konjce, Slovenia) was run at 58 °C during the assay (up to 180 days) and the percentage of biodegradation (D_t_) was collected by calculating the maximum theoretical amount of CO_2_ (determined by measuring the total organic carbon (TOC) content) and measuring the CO_2_ generated inside the bioreactors according to the equations proposed in international standards (1) and (2):
(1)$$Th{CO}_{2}=M_{TOT}\times C_{TOT}\times \frac{44}{12}$$ where *M*_TOT_ represents the total dry solids, in grams, in the test material introduced into the composting containers at the beginning of the test; *C*_TOT_ signifies the ratio of total organic carbon in the total dry solids in the test material, in grams per gram; and 44 and 12 denote the molecular mass of carbon dioxide and the atomic mass of carbon, respectively.
(2)$$D_{t}=\frac{{{(CO}_{2})}_{t\;}-{{(CO}_{2})}_{B\;}}{Th{CO}_{2}}\times 100$$ where (CO_2_)_T_ signifies the accumulated amount of carbon dioxide generated in each composting container containing the test material, in grams per container; (CO_2_)_B_ denotes the average accumulated amount of carbon dioxide generated in the blank containers, in grams per container; and ThCO_2_ represents is the theoretical amount of carbon dioxide that the test material can produce, in grams per container.

For the bioreactors (3 L of capacity), a 3-month-old compost made from manure and straw prepared at the EPSO (Escuela Politécnica Superior de Orihuela, Miguel Hernández University) was used as the inoculum, and perlite and mature compost were used as the substrate. The compost mixture was sieved using a 10 mm mesh, and deionized water was added to increase the moisture content to approximately 50%, facilitated by the OHAUS MB23 (OHAUS Corporation, Parsippany, NJ, USA) thermobalance humidity analyser. A laboratory-scale composting process tailored to the materials under study was developed, wherein collagen powder from Sigma-Aldrich^®^ (St. Louis, MO, USA) was used as a positive reference control substance, as it is the main component of leather.

### 2.2. Characterization

The physicochemical properties of the leather samples were analyzed, including the heavy metal content, moisture, pH, total organic matter (TOM), total carbon (TC), total nitrogen (TN), and the C/N ratio [15], in order to obtain the necessary information to prepare an adequate initial mixture for composting (10–40 C/N) and calculate the maximum theoretical amount of CO_2_. The pH was analyzed in a 1:10 (*w*/*v*) sample/water solution. The TC and TN were determined via dry combustion at 1020 °C in an automatic microanalyzer (Thermo Finnigan Flash 1112 Series, Waltham, MA, USA). The TOM was evaluated by the loss on ignition after the incineration of dry samples at 550 °C for 4 h. The concentration of metals (Zn, Cu, Ni, Pb, Cd, Mg, Mn, As, Se, and Cr) was determined using inductively coupled plasma mass spectrometry (7800 ICP-MS, Agilent, Santa Clara, CA, USA) according to the CPSC-CH-E1002-08.03 method for metal content determination. Additionally, fluorine was determined using ionic chromatography using a Metrohm 930 Compact IC Flex (Herisau, Switzerland).

The maturity of the compost obtained was assessed through studies of its chemical properties. Furthermore, ecotoxicity tests were conducted according to the germination index [16], combining the measurements of seed germination and root elongation, for cress seeds (*Lepidium sativum* L.), according to the international standard OECD 208 [17] for *Raphanus sativus* L. A one-way analysis of variance (ANOVA) of the mean values for each maturity was used in order to test the statistically significant differences across the different treatments.

### 2.3. Spectrometry Analysis

The behavior of microorganisms and their interactions with finished leather and degraded materials were evaluated spectrometrically. Samples were characterized using advanced analytical techniques, such as FT-IR, TGA, and ^13^C CP-MAS NMR. FT-IR spectroscopy is an analytical technique widely used in the study of the biodegradation of plastics and polymeric materials. This technique allows for the analysis of the chemical and structural changes that occur in polymers during the biodegradation process. FT-IR spectroscopy is based on the interaction of infrared radiation with the molecules in a sample. Using this technique, a spectrum is obtained showing the different vibrations of the molecules present in the sample. These vibrations are characteristic of the functional groups present in plastics and other polymers and can provide information about the chemical composition and structural changes [18,19,20,21,22,23,24,25,26]. The FT-IR spectra were collected using a Bruker IFS 66 spectrometer (Billerica, MA, USA) using the ATR technique. The resolution was set at 4 cm^−1^, and the operating range was 400–4000 cm^−1^. For TGA, samples were air-dried, ground in an agate mill, sieved through a 0.125 mm mesh, and milled again using a mortar, with liquid N_2_ [27].

Thermogravimetry is an analytical technique used to study changes in the mass of a material as a function of the temperature [28,29]. It is based on the precise measurement of the mass variation in a sample while it is subjected to a controlled heating program. As the temperature increases, changes in the mass of the sample are recorded, providing information on the processes of thermal decomposition, oxidation, dehydration, sublimation, and other related phenomena. Depending on the objective of the study, thermogravimetry can be performed under different atmospheric conditions. For example, it can be carried out in air, nitrogen, a vacuum, or other controlled atmospheres. The choice of atmosphere depends on the nature of the sample and the process to be studied. In this case, an inert atmosphere was used. The assays were performed using a TGA/SDTA851e/LF/1600 (Mettler Toledo, Barcelona, Spain). All the samples were degraded with inert N_2_ gas in a temperature range from 25 to 700 °C using a heating rate of 10 °C min^−1^, a sample weight of about 25 mg, an Al_2_O_3_ pan, and self-controlled calibration.

Cross-polarization/magic angle spinning nuclear magnetic resonance (CP-MAS NMR) is a technique used to study the structure and dynamics of solid materials at the atomic level [30,31,32,33,34]. Solids and polymeric materials present additional challenges due to the lack of molecular mobility and the presence of inhomogeneous dipolar interactions. By combining CP-NMR and MAS, it is possible to overcome the challenges associated with the lack of order and mobility in solids by providing detailed information on the atomic and molecular structure. The ^13^C-CPMAS NMR experiments were performed using a Bruker Advance DRX500 spectrometer (Billerica, MA, USA) operating at 125.75 MHz for ^13^C. Samples were packed into a 4 mm diameter cylindrical zirconia rotor with Kel-F end-caps and spun at 2000 ± 10 Hz. A conventional CPMAS pulse sequence was used [35], with a contact time of 1.0 ms. A total of 4000 scans were accumulated, with a pulse delay of 1.5 s. The line broadening was adjusted to 50 Hz. The spectral distributions (the distribution of total signal intensity among various chemical shift ranges) were calculated in seven chemical shift regions: carbonyl (210–165 ppm), O (aromatic) (165–145 ppm), aromatic (145–110 ppm), O_2_-alkyl (110–95 ppm), O-alkyl (95–60 ppm), N-alkyl/methoxy (60–45 ppm), and alkyl (45 to −10 ppm) [36]. The labels indicate only the major types of C found in each region.

### 2.4. Samples

This study aimed to determine the specific influence of different types of currently used finishing chemicals, with varied chemical natures, on the biodegradation properties of leather, as can be seen summarized in Table 1. Additionally, bio-based finishing products from renewable sources, indicated by BIO, were used. These include crosslinking agents based on polymers and resins (IS, EA, and NL), which are known for their high toxicity; bio-based materials with aqueous finishes (AB, PTB, and PFB); widely used protein-based products such as binders and adhesives (CAS and EB); and finishing and treatment agents (EX and BP) such as wax and metal-based pigments.

The composition and degradability of the different materials were investigated following the determination of their elemental composition (Table 2). This analysis enabled the calculation of the C/N ratio for each material and the estimation of the theoretical amount of CO_2_ deposited in each bioreactor. A uniform quantity of material (±20 g) was employed across the biodegradation assay. Compostability testing based on the C/N ratios was deemed impractical due to the challenges associated with adjusting the diverse C/N values to a common range without introducing additives, as well as the limited capacity of the bioreactors.

Table 3 shows the amounts of heavy metals in the examined materials. The material groups were divided into batches, thereby simplifying the analyses to be conducted, focusing solely on the batches and metals that exhibited high quantities of any of the elements presented (Table 3). As shown, in batch C, AB displayed a high amount of Zn (1328.6) owing to its composition based on vegetable oils and cellulose. It can be employed as an additive to enhance the mechanical strength, thermal conductivity, and antibacterial properties of the material. The remainder of the materials examined did not contain notable amounts of heavy metals; the amount generally fell below the detection limit, and therefore, the materials’ biodegradability was not affected.

## 3. Results and Discussion

### 3.1. Biodegradation and Composting

The presented results (Table 4) facilitate a detailed analysis of the biodegradability of various leather-finishing products according to the ISO 14855-1 standard [1]. Significant variability was observed in the biodegradability percentages of both individual finishing products and when applied to CRT. PTB and PFB exhibited high biodegradability when combined with leather, with values of 79.37%, which highlighted their potential as sustainable finishes. In this regard, the EA, EX, and BP products stood out for their high biodegradability (they did not interfere with the microbiota), with percentages exceeding 80% when applied to leather.

In contrast, the IS, NL, and CAS products showed moderate biodegradability, ranging between 50% and 60% (Table 4), due to collagen degradation in the crust leather. Regarding the latter product (CAS), the decomposition of organic materials with a low C/N ratio (Table 2) can produce a series of adverse effects: the excessive release of ammonia during composting via volatilization can produce an inhibition and imbalance in the microbiota due to its toxicity; and the acidification of the compost due to the excessive production of organic acids via protein decomposition, as noted by other authors [37]. As a result, the AB option requires further investigation, as it showed a low biodegradability (58.38%) due to its chemical structure, which has the same complexity as that of the base product that it replaced and exhibits the same functional properties. Additionally, as previously observed, it contains significant amounts of zinc in its formulation, which confers antibacterial and antimicrobial properties [38] and affects the biodegradation rates.

### 3.2. Maturity

The germination index (GI) [39] is considered the most comprehensive indicator for describing the phytotoxic potential of an organic material, as it incorporates radicle elongation, allowing the examination of substrates that, while not preventing germination, may limit root development and consequently the development of the future adult plant [40,41,42]. This index, together with the OECD 208 ecotoxicity test, allows for the evaluation and analysis of the maturity of the compost produced from leathers impregnated with different finishing products based on four variables: the relative germination percentage (RGeP), total biomass ratio (*Biomass R*), relative growth percentage (RGrP), and germination index (GI).

For each of the four parameters (RGeP), the effect of using each of the 11 samples was analyzed. The results obtained from each of the 10 plates used in the tests were utilized for this analysis. An analysis of variance procedure was used. The samples that showed significant differences from the others are discussed in the text. To compare the results between samples, a multiple comparison procedure with the Bonferroni correction was used.

The relative germination percentage (RGeP) showed that most treatments hovered around 100%, indicating effective germination, similar to the controls, with notable exceptions, such as the PTB and PFB materials (63.9%), suggesting that the possible chemical migration of the products into the compost matrix affected seed germination, although it did not prevent their biodegradation. Regarding *Biomass R*, the high variability of the observed data indicated a weak correlation between this parameter and the others listed (Table 5), suggesting that the total biomass might be influenced by other factors not directly measured by these variables. In this context, as an outstanding product, it was observed that CAS significantly improved the performance (151.5) of the control culture due to this compound belonging to the group of phosphoproteins and being rich in amino acids that stimulate plant metabolism under limited nitrogen conditions, as observed in in vitro cultures of *Nicotiana Tabacum* sp. [43]. However, it did not show the same behavior for the plant growth parameter (RGrP), where it was one of the treatments that developed the shortest growth, proving to be an immature compost, as we can see from the GI parameter. In this regard, the products composed of BP, EA, and EX stood out, with values of 125.9%, 123.1%, and 114.4%, respectively.

The GI, which integrates variables such as the RGeP and RGrP, shows the bio-stimulant power of the plant metabolism of the base minerals in BP and EX, while the low GI of PTB and PFB corroborated the previous discussion regarding the RGeP of these polyurethane products and indicated the presence of phytotoxic compounds, as shown by the germination index values obtained (GI ≤ 60%) compared to a control with distilled water (Table 5). It is also interesting to note that the product formed by CAS showed values indicating immature compost, as suggested by the incomplete biodegradation (62.14%) and the possible presence of ammoniacal compounds and the acidification of the medium. The rest of the studied composts showed the absence of phytotoxicity in the finishing products and an optimal degree of maturity.

### 3.3. Spectroscopy and Thermal Analysis

The decomposition process of a series of individual finishing products used in the footwear industry was evaluated. These materials were individually subjected to a biological decomposition process of composting (58 °C), using perlite as an inert material during the process. The samples obtained at the end of the composting process were analyzed via FT-IR spectroscopy, which allowed the evaluation of the presence of the functional chemical groups that formed the different materials. It was observed that some samples presented the same spectrum before and after the biological degradation process, which indicated that they had not been altered (Figure 1A). When the sample had fragmented, it was not always possible to separate its signals from those of the perlite, so these signals were observed together with signals from the starting material (Figure 1A). In other cases, the material had not been altered at all and it was easy to separate it from the perlite (Figure 1A).

With the intention of using the spectrum as a fingerprint for the degradation process of the composted material, it was also compared to the solid-state NMR spectra and the profiles obtained from the thermal decomposition of the materials (Figure 1).

The first material studied was IS. The spectra of the sample before and after composting were very similar. The differences in the FT-IR spectra were due to the presence of residual water in the perlite remains; that is, the signals from the initial material in the FT-IR spectra were the same at the beginning and at the end of the process. Based on the thermal analysis profile, it was observed that they were practically identical. The same occurred in the CP-MAS spectrum (Figure 1B), where the signals were the same. There was some difference due to the presence of perlite together with the starting material. The presence of perlite increased the noise in the spectrum, as the amount of polymeric material was diluted with the accompanying perlite. The most intense signal in the DTG profile indicated that most of the material corresponded to aliphatic C chains, which are very resistant, showing that the material degrades at temperatures close to 500 °C. This was confirmed by the solid NMR spectrum (Figure 1B), in which the most intense signal (between 20 and 30 ppm) corresponded to this type of C. The signal around 48 ppm corresponds to C=CH_2_C. The signal around 72 ppm corresponds to C bound to -OH groups, although it was a very minor signal. According to the various techniques used, it can be concluded that this material had not been altered during the bacterial degradation process to which it had been subjected. These polymeric materials are not easily degraded by microorganisms, as confirmed by the results (Figure 1).

The next sample, EA, was not altered during the composting process, as can be seen in Figure 2A. The FT-IR spectra were practically the same before and after composting. The same was true for the thermal analysis profile (Figure 2A). As with the IS sample, the CP-MAS spectrum (Figure 2B) showed the presence of aliphatic chains in the region between 14 and 50 ppm, which are difficult to degrade. This meant that the material remained unaltered.

Based on the FT-IR spectra (Figure 3A), the sample corresponding to NL does not seem to have undergone significant alterations, although it can be observed that changes occurred in certain signals when observing the DTG profiles (Figure 3A) and the CP-MAS spectra (Figure 3B). The CP-MAS spectrum of the final sample contained weak signals, as the signal-to-noise ratio was not very high. Narrow signals were observed at around 70 ppm, which corresponded to the C-OH, carbohydrates, and O acyl ether groups that are present in cellulose- and hemicellulose-type rings. Other weak signals were observed in the alkyl C region (30–20 ppm) (Figure 3B). The signals at around 70 ppm decreased during the composting process, indicating partial degradation of the material. The analysis of the spectra indicates a partial degradation of NL. From the FT-IR spectra, it was difficult to identify this degradation of the cellulose, since the signal for the C-OH group appeared at around 1100 cm^−1^ and in the same region as the signal for the silicates of the perlite. However, it was possible to identify not only the degradation of the polymeric material, but also whether degradation of some part of the polymer composition itself was occurring, as was the case here.

The next sample analyzed was AB. This is a material rich in aliphatic chains, as can be seen from the CP-MAS spectrum (Figure 4B). In the thermal analysis profile and in the FT-IR spectra, it was not possible to observe changes in the signals of the functional groups of the material (Figure 4A). Very few changes were observed in the absorption bands, except at 1631 (vibration C=C (alkenes and aromatics) and vibration C=O (of amides)) and 3377 cm^−1^ (vibration O-H) (Figure 4A). These bands corresponded to -OH groups linked to the presence of residual water in the perlite substrate [44].

In this sample, a very intense signal was also observed in the DTG profile, which was also associated with the presence of an aliphatic-type polymer, which degraded at high temperatures (between 400 and 500 °C) (Figure 4A), although in this sample, there were also C-OH groups. This can be seen from the solid NMR spectrum, in which the most intense and sharpest signal appeared at 20 ppm, which corresponded to aliphatic carbon. The signal at around 48 ppm corresponds to C=CH_2_C. The signal at around 72 ppm corresponds to C attached to -OH groups (Figure 4B). The absence of the changes in the signals of the FT-IR and CP-MAS spectra and in the DTG image indicates that the sample remained practically unchanged.

The CAS sample, on the contrary, completely degraded during the composting process. In the FT-IR spectra, only the characteristic signals of the perlite substrate appeared (Figure 5A). In the NMR spectra, at the end of the composting process, no signal was observed, which indicated the complete degradation of the material (Figure 5B). Casein is a protein present in milk and is rich in phosphate and calcium groups. It is an easily degradable protein, as can be seen from the FT-IR and CP-MAS spectra and the DTG profile (Figure 5). The bands at around 1630 and 3294 cm^−1^ correspond to the residual water adsorbed on the perlite (Figure 5A), which appeared at 992 cm^−1^ [45].

The next material, EB, has a casein base, as can be seen from the initial FT-IR and CP-MAS spectra (Figure 6). At the end of the composting process, this protein base had completely degraded, leaving the remains of aliphatic chains (Figure 6B).

This sample, EB, had indeed degraded during the biological treatment of the sample, leaving only the perlite support with residual water molecules. The signals of aliphatic C (at around 2900 cm^−1^) were also observed (Figure 6A). These aliphatic C signals could be due to the additive that accompanied the protein base, which has not biodegraded, although these possible traces were not seen in either the CP-MAS spectrum or the DTG profile (Figure 6). In the initial NMR spectrum of the sample (blue), a group of broad signals was observed between 50 and 20 ppm, corresponding to various types of aliphatic chains and C=CH_2_C, which almost disappeared. There were also some broad signals at around 50–60 ppm, corresponding to N-CH_2_ or N-CH_3_. At around 140 ppm, there was a signal corresponding to aromatic C=C bonds. At 175 ppm, there was a signal corresponding to aromatic C-O bonds. Above 200 ppm, C=O groups were observed, corresponding to ketones and aldehydes. All these signals disappeared at the end of the biological degradation process of the starting material, as can be observed in the CP-MAS spectra (Figure 6B).

The next sample, EX, appeared to have partially degraded. In the FT-IR spectrum (Figure 7A), it was only possible to identify aliphatic C signals in the sample after the composting process, and the same was true for the CP-MAS spectrum. This sample probably needed a slightly longer time to completely degrade.

The EX sample was rich in aliphatic chains (Figure 7A) and almost totally degraded during the biological treatment, as can be seen in the spectra; however, there were still some undegraded aliphatic chains present, as can be deduced from the CP-MAS spectrum, with a very clear signal at around 30 ppm, and from the FT-IR spectrum. In the DTG profile, a small shoulder is observed at around 500 °C, where the material degraded, which corresponds to the remains of aliphatic chains (Figure 7A).

The PTB and PFB materials were also analyzed.

The results for these samples (Figure 8 and Figure 9) showed that there had been no degradation in the material. As with many previous samples, a narrow signal was observed between 40 and 30 ppm, corresponding to various types of aliphatic chains and C=CH_2_C. A broad double signal was also observed at above 50 ppm and was associated with N-CH_2_ or N-CH_3_. There were also some narrow signals at around 70 ppm, corresponding to C-OH, carbohydrates, and O acyl ether groups that are present in cellulose- and hemicellulose-type rings. Signals were observed in the aromatic region; at around 160 ppm, an aromatic C-O signal was observed, but it was of very low intensity.

As for C+, a protein such as collagen will evidently decompose, as proven by the degradation during the biological treatment (Figure 10), since microorganisms are strongly dependent on protein nitrogen and collagen degrades via biological action to supply nitrogen to different microorganisms. Again, in the FT-IR spectrum, bands were observed at around 1630 (vibration C=C (alkenes and aromatics) and vibration C=O (of amides)) and 3294 cm^−1^ (vibration O-H), corresponding to the water molecules absorbed on the perlite, which appeared as a band at 992 cm^−1^ (vibration Si-O) (Figure 10A). In the CP-MAS spectrum, as well as in the DTG profile, no collagen signal was observed, meaning that it had completely degraded during the biological treatment (Figure 10). In the DTG profile, the peak of the thermal degradation of the proteins in the initial sample was observed at around 300 °C. In the final sample, there was no sign of the degradation of organic compounds; only a peak from water evaporation (100 °C) was observed (Figure 10A).

The NP sample was difficult to analyze due to its nature as a carbon black base. The FT-IR spectra for the initial sample did not provide much information (Figure 11), although none of the materials appeared in the final sample. The same occurred in the DTG profiles, where it was observed that the signals of the initial materials had disappeared (inset in Figure 11). Using CP-MAS, no signals were seen either in the initial or in the final material. This is due to the fact that the technique used, CP-MAS, requires C-H atoms, while a carbon black is mainly formed by quaternary carbons, without H atoms.

The BP pigment sample had degraded almost completely, although traces were still visible in the DTG profile (inset in Figure 11). The final sample also presented a black color, characteristic of the initial material, so it could be deduced that there was still undegraded material, although a small amount remained at the end of the composting process.

## 4. Discussion and Conclusions

An analysis of the biodegradability of leather-finishing products were undertaken to underscore the importance of selecting sustainable options in leather manufacturing. Protein-based materials (CAS, EB, and C+ as a control) (Figure 5, Figure 6 and Figure 10) exhibited complete biodegradation during the studied composting processes. These materials are readily degraded by proteases produced by microorganisms [46,47,48]. They serve as an excellent source of assimilable nitrogen for the microorganisms involved in the biodegradation process [49,50]. The biodegradation of these materials can be observed through FT-IR spectroscopy and thermal analysis, which, in conjunction with CP-MAS spectroscopy, confirm the complete degradation of protein-based materials [47,51,52].

Other materials, such as those based on nitrocellulose or wax (NL and EX), exhibit partial degradation of their molecular structures (Figure 3 and Figure 7). During composting processes, the degradation of cellulosic and nitrocellulosic materials of different compositions has been analyzed [32,46,53,54]. This degradation has also been observed in waxes, although there are few references in the literature regarding the degradation of these materials [55]. We found that wax-based materials are proposed as biodegradable, although there are no scientific studies to support these claims. Our work suggests that degradation is occurring; however, a longer period may be needed for complete degradation due to their hydrophobic nature and rich aliphatic bonds.

Materials based on acrylic compounds (EA and AB) (Figure 2 and Figure 4) remained unaltered during the studied biodegradation process. These materials are highly resistant to microbial degradation and are present as microplastics in various natural environments as well as in compost itself, acting as a vehicle for their presence in the environment [51,52,53,54,55].

An analysis of a sample based on IS was also undertaken (Figure 1). This sample also remained unaltered during the biodegradation process. Isocyanates are key compounds in the synthesis of polyurethanes, which we also studied in our work. We evaluated the biodegradation of two types of polyurethanes (PTB and PFB) (Figure 8 and Figure 9). These materials showed no alterations when analyzed with the techniques used in this study (FT-IR, DTG, and CP-MAS). Considering the enormous variety and structural diversity of these polymers, there are descriptions in the literature [56,57,58,59] of the biodegradation of other polyurethanes, although with fairly long composting times.

The combination of various spectroscopic techniques (FT-IR and CP-MAS) and thermal analysis (DTG) allows us to obtain a fingerprint of the materials to evaluate their biodegradation under controlled conditions [22,23,60]. We would like to highlight the use of CP-MAS, as it enabled us to evaluate the materials mixed with the substrate used (perlite), since the technique is based on detecting carbon-based compounds and perlite is composed of silicates. This material does not degrade during thermal analysis, nor does it generate any signal in CP-MAS, only presenting a broad vibration band of the Si-O-M group (M=Al or Si) at around 1000 cm^−1^ [45].

In this study, the germination index (GI), along with the OECD 208 ecotoxicity test, effectively integrated variables such as the relative germination percentage (RGeP) and the relative growth percentage (RGrP), providing a comprehensive view of compost maturity and potential phytotoxic effects. In this regard, the absence of phytotoxicity and the optimal maturity observed in most of the studied composts highlight the feasibility of using certain leather-finishing products in sustainable compost production. However, the phytotoxic effects observed in specific treatments (e.g., PTB and PFB) underscore the importance of understanding the composition and management of composting processes to mitigate any potential phytotoxic outcomes. Research on the degradation of MDI-based polyurethane under composting conditions revealed the release of 4,4′-diaminodiphenylmethane (MDA), a hazardous by-product. This indicated potential phytotoxicity [61] and the need for a thorough evaluation of finishing products to ensure they do not negatively impact compost quality and plant health [62]. Additionally, a broader range of plant species should be tested to evaluate the generalizability of the results, as different plants might respond differently to the produced compost [63]. The findings of this work demonstrate the complex interactions between polyurethane degradation products and plant growth, emphasizing the need for further research to fully understand the phytotoxic effects of compost containing polyurethane [64].

The results obtained in this study suggest that the choice of finishing product has a considerable impact on the final biodegradability of leather, positioning EA-based and animal-derived options as more sustainable alternatives compared to other, chemically synthesized options, such as IS or NL. This conclusion has also been reached in similar studies [65] on the laboratory-scale composting of leathers with different tanning processes. In other works where the compostability of leathers with different finishes was studied [66,67], a multidisciplinary approach was not adopted to individually evaluate the main finishing products used in the leather industry, nor were the chemical changes produced on the leather surface as a result of aerobic thermal treatment during composting or the final ecotoxicity of the composts studied as a parameter of maturity. This work addresses the limitations of other studies to delve into the chemical matrix that composes the finishing products used. However, we consider it necessary to approach this on a different scale than the proposed laboratory composting scale, such as a pre-industrial or industrial composting process, as some studies developed on an industrial scale used other tanning cycle materials, such as tannery sludges [68].

## Figures and Tables

**Figure 1 polymers-16-01908-f001:**
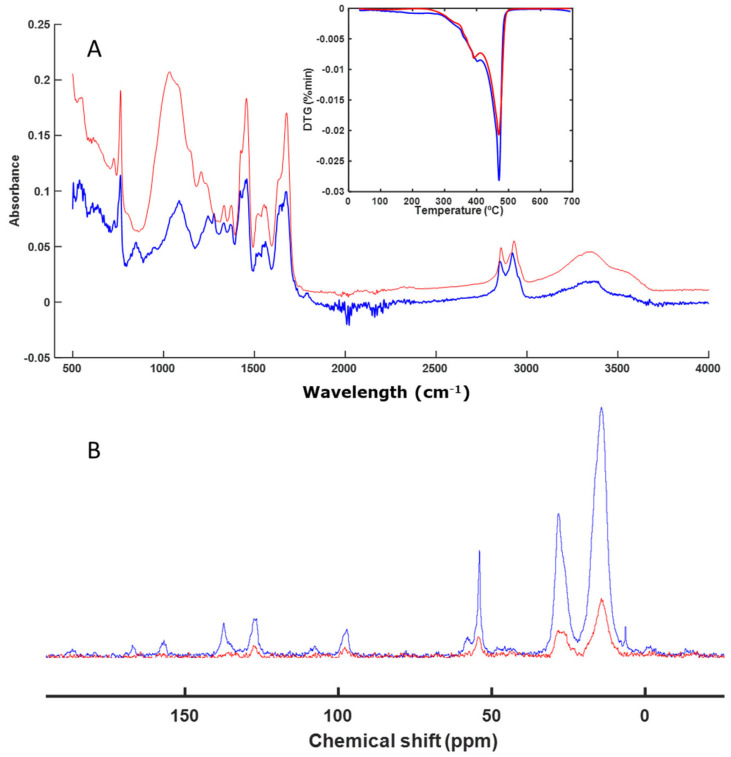
ATR–FT–IR with DTG profile (inset) (**A**) and CP-MAS spectra (**B**) of IS. The spectrum of the initial sample is shown in blue, and the spectrum of the final sample is shown in red.

**Figure 2 polymers-16-01908-f002:**
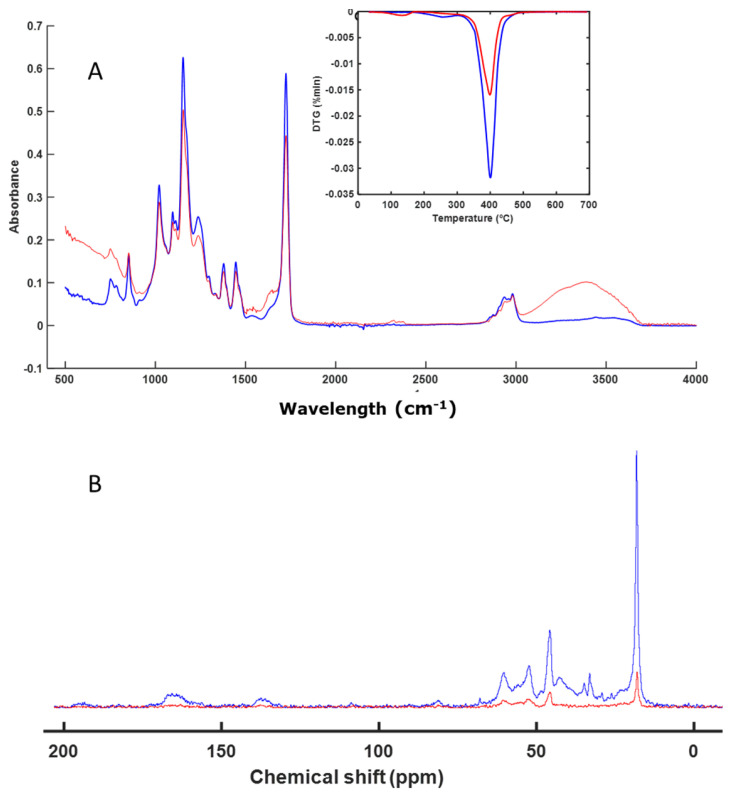
ATR–FT–IR spectra with DTG profile (inset) (**A**) and CP-MAS spectra (**B**) of EA. The spectrum of the initial sample is shown in blue, and the spectrum of the final sample is shown in red.

**Figure 3 polymers-16-01908-f003:**
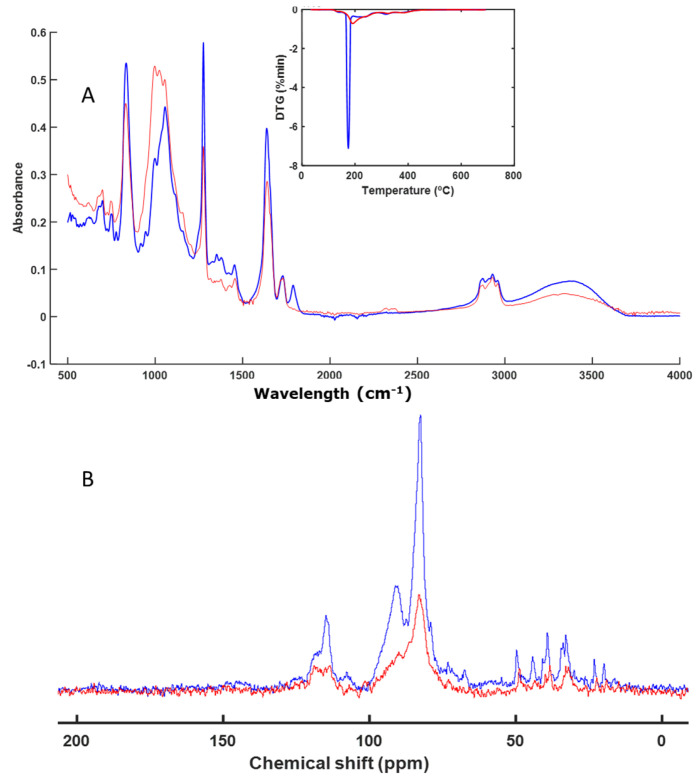
ATR–FT–IR spectra with DTG profile (inset) (**A**) and CP-MAS spectra (**B**) of NL. The spectrum of the initial sample is shown in blue, and the spectrum of the final sample is shown in red.

**Figure 4 polymers-16-01908-f004:**
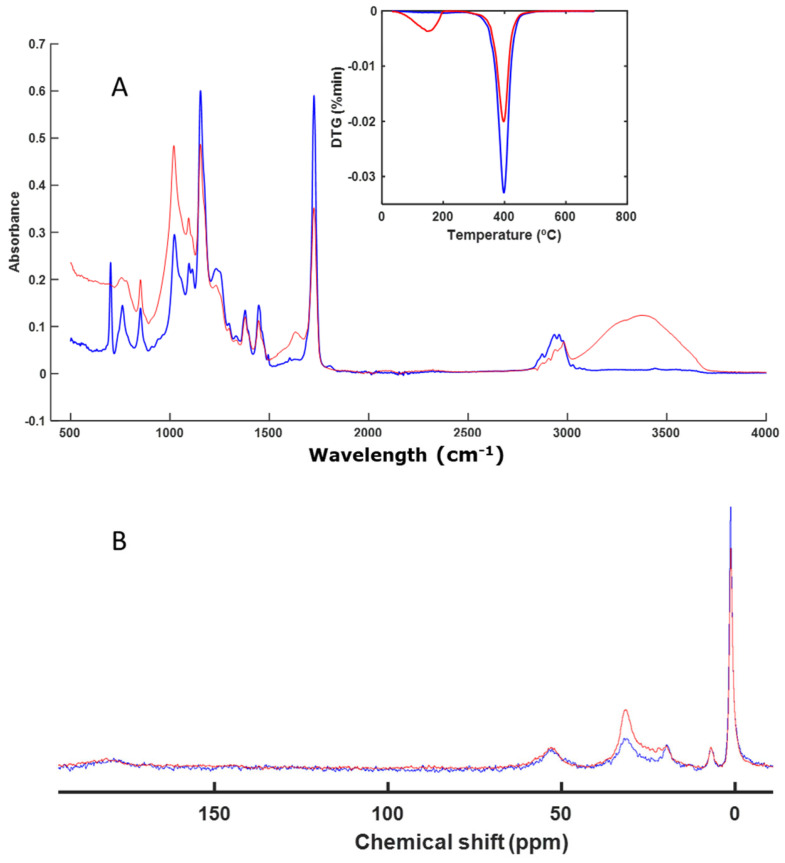
ATR–FT–IR spectra with DTG profile (inset) (**A**) and CP-MAS spectra (**B**) of AB. The spectrum of the initial sample is shown in blue, and the spectrum of the final sample is shown in red.

**Figure 5 polymers-16-01908-f005:**
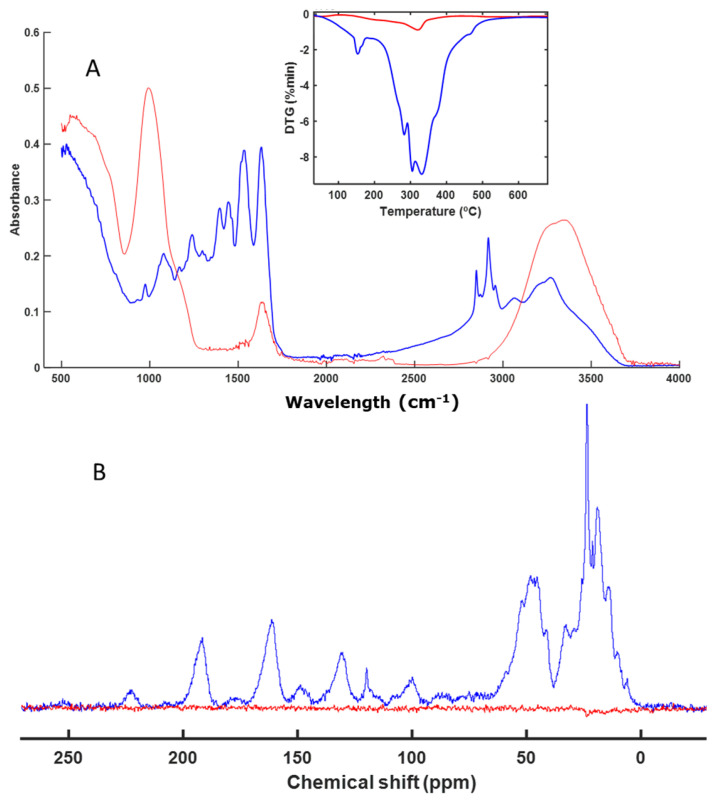
ATR–FT–IR spectra with DTG profile (inset) (**A**) and CP-MAS spectra (**B**) of CAS. The spectrum of the initial sample is shown in blue, and the spectrum of the final sample is shown in red.

**Figure 6 polymers-16-01908-f006:**
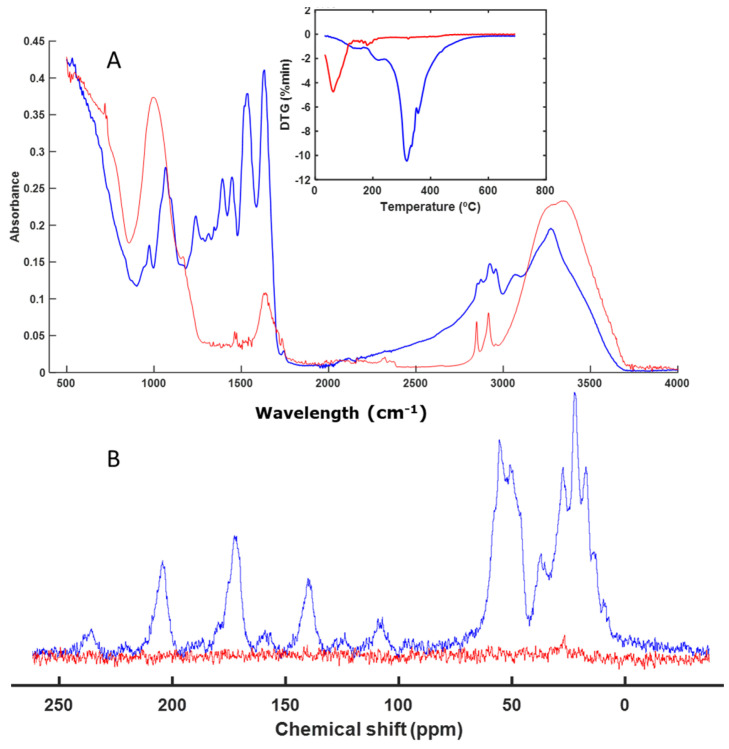
ATR–FT–IR spectra with DTG profile (inset) (**A**) and CP-MAS spectra (**B**) of EB. The spectrum of the initial sample is shown in blue, and the spectrum of the final sample is shown in red.

**Figure 7 polymers-16-01908-f007:**
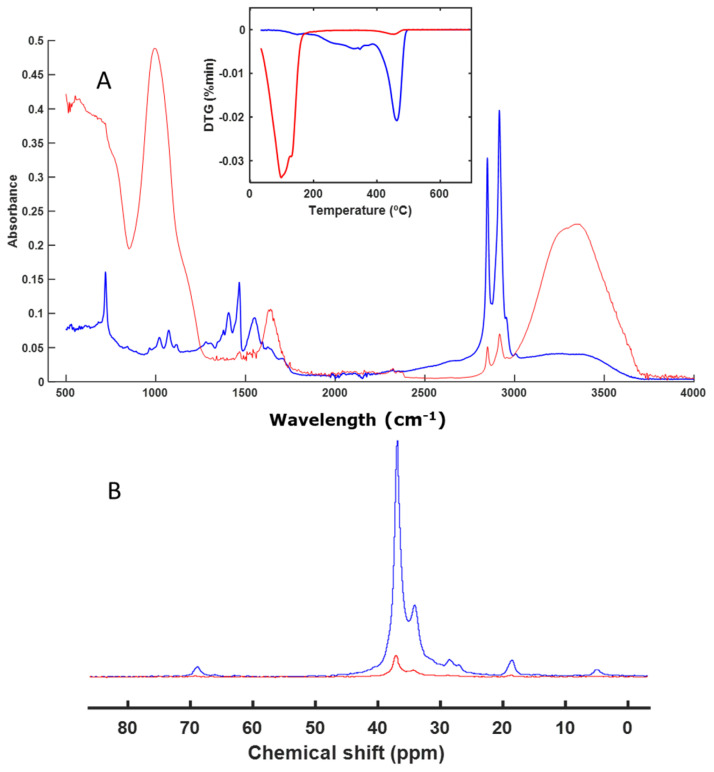
ATR–FT–IR spectra with DTG profile (inset) (**A**) and CP-MAS spectra (**B**) of EX. The spectrum of the initial sample is shown in blue, and the spectrum of the final sample is shown in red.

**Figure 8 polymers-16-01908-f008:**
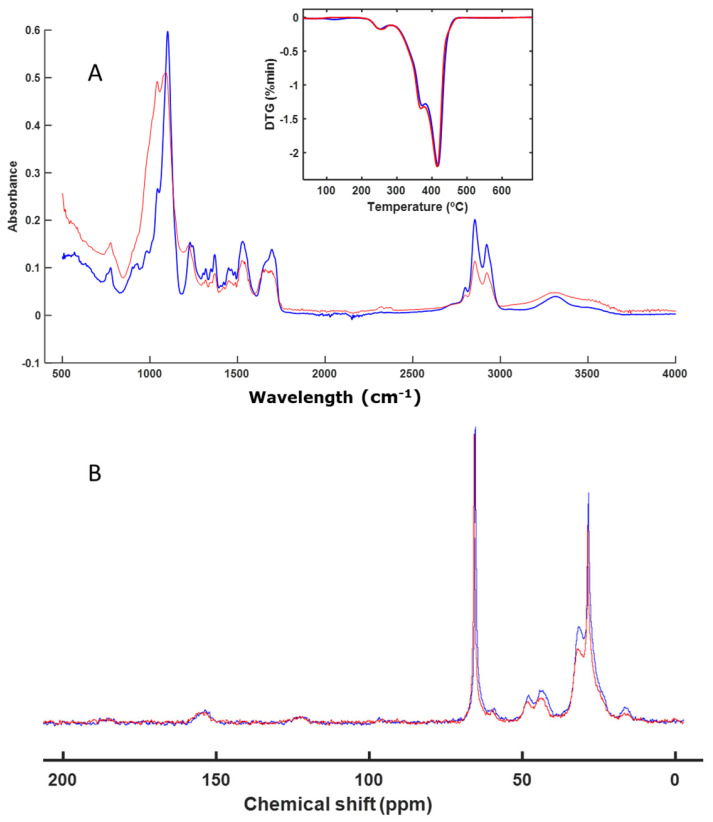
ATR–FT–IR spectra with DTG profile (inset) (**A**) and CP-MAS spectra (**B**) of PTB. The spectrum of the initial sample is shown in blue, and the spectrum of the final sample is shown in red.

**Figure 9 polymers-16-01908-f009:**
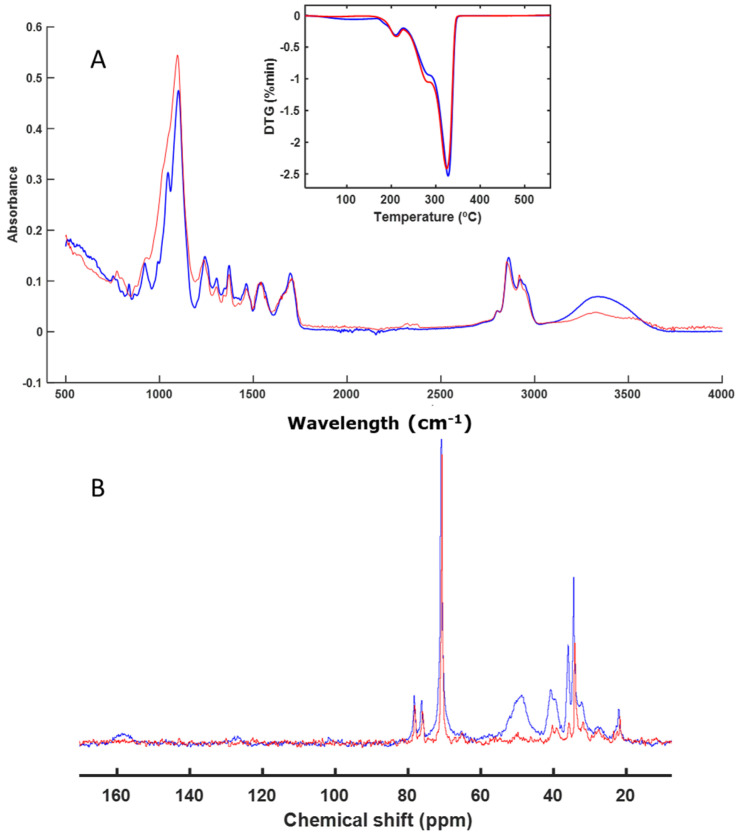
ATR–FT–IR spectra with DTG profile (inset) (**A**) and CP-MAS spectra (**B**) of PFB. The spectrum of the initial sample is shown in blue, and the spectrum of the final sample is shown in red.

**Figure 10 polymers-16-01908-f010:**
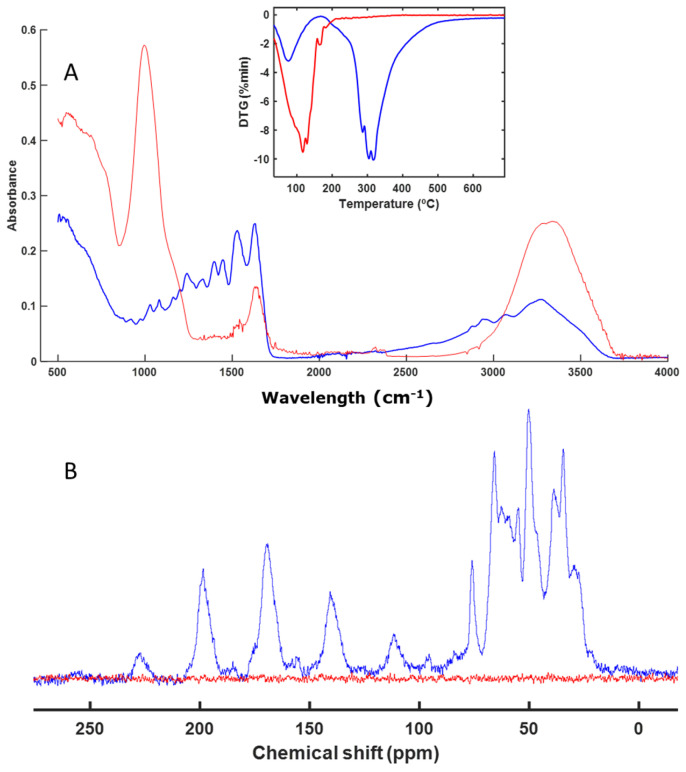
ATR–FT–IR spectra with DTG profile (inset) (**A**) and CP-MAS spectra (**B**) of C+. The spectrum of the initial sample is shown in blue, and the spectrum of the final sample is shown in red.

**Figure 11 polymers-16-01908-f011:**
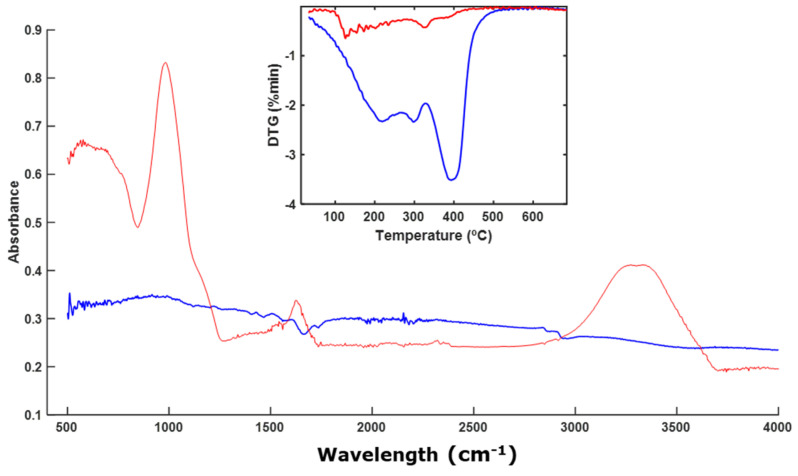
FT–IR spectrum with the ATR of BP. The spectrum of the initial sample is shown in blue, and the spectrum of the final sample is shown in red. The DTG profile is inset.

**Table 1 polymers-16-01908-t001:** Finishing products used.

Groups	Sample	Code	Leather
Polymers and resins	Isocyanate	IS	Crust leather + IS
	Acrylic	EA	Crust leather + EA
	Nitrocellulose lacquer	NL	Crust leather + NL
Bio-based materials	Acrylic BIO	AB	Crust leather + AB
	Polyurethane top BIO	PTB	Crust leather + PTB + PFB *
	Polyurethane primer BIO	PFB	Crust leather + PTB + PFB *
Binders and adhesives	Casein	CAS	Crust leather + CAS
	Protein binder	EB	Crust leather + EB
Finishing and treatment agents	Wax	EX	Crust leather + EX
	Black pigment	BP	Crust leather + BP
Control	Collagen	C+	Collagen
	Crust	CRT	Crust leather

* Composite sample in the same bioreactor.

**Table 2 polymers-16-01908-t002:** Elemental analysis results for the studied materials.

Sample	Code	Carbon (%)	CO_2_ (g)	Nitrogen (%)	C/N
Polymers and resins					
Isocyanate	IS	60.42	20.49	15.85	3.81
Acrylic	EA	59.08	20.58	0.83	71.35
Nitrocellulose lacquer	NL	44.09	20.21	6.83	6.45
Bio-based materials					
Acrylic BIO	AB	57.59	20.59	0.72	79.92
Polyurethane top BIO	PTB	61.63	20.34	5.10	12.10
Polyurethane primer BIO	PFB	98.35	20.74	0.23	433.4
Binders and adhesives					
Casein	CAS	51.21	20.65	12.24	4.19
Protein binder	EB	47.19	20.76	13.45	3.51
Finishing and treatment agents					
Wax	EX	79.96	20.52	1.46	54.87
Black pigment	BP	71.28	20.91	3.85	18.49
Control					
Collagen	C+	45.39	20.21	16.93	2.68
Crust	CRT	47.93	--	10.49	4.57

**Table 3 polymers-16-01908-t003:** Heavy metal amounts in the studied materials (mg/kg).

Sample	Code	Cr	Ni	Cu	Zn	As	Se	Mo	Cd	Pb	Hg
Batch A	EA	<1	<1	<5	<1	<1	<1	<1	<1	1.2	<0.15
	EX	<1	<1	<5	<1	<1	<1	<1	<1	1.2	<0.15
	EB	<1	<1	<5	<1	<1	<1	<1	<1	1.2	<0.15
Batch B	PTB	<1	<1	<5	<1	<1	<1	<1	<1	<1	<0.15
	PFB	<1	<1	<5	<1	<1	<1	<1	<1	<1	<0.15
Batch C	CAS	<1	<1	5.6	<1	<1	<1	<1	<1	1	<0.15
	BP	<1	<1	5.6	<1	<1	<1	<1	<1	1	<0.15
	AB	<1	<1	5.6	1328.6	<1	<1	<1	<1	1	<0.15
Batch D	IS	<1	<1	<5	<1	<1	<1	<1	<1	<1	<0.15
	NL	<1	<1	<5	<1	<1	<1	<1	<1	<1	<0.15
Control	C+	<1	<1	<5	<1	<1	<1	<1	<1	<1	<0.15
	CRT	<1	<1	<5	<1	<1	<1	<1	<1	<1	<0.15

**Table 4 polymers-16-01908-t004:** Biodegradability values (%) of finishing products and of leather + each finishing product.

Sample	Finishing Product ThCO_2_ (g)	Biodegradability %	Leather + Finishing Product ThCO_2_ (g)	Biodegradability %
IS	20.49	13.83	18.33	57.30
EA	20.58	0.00	19.19	84.20
NL	20.21	5.73	18.84	52.91
AB	20.59	3.04	18.30	58.38
PTB	20.34	38.25	19.16 *	79.37 *
PFB	20.74	0.00	19.16 *	79.37 *
CAS	20.65	87.23	18.17	62.14
EB	20.76	5.48	19.37	82.53
EX	20.52	60.1	19.42	85.09
BP	20.91	26.27	19.59	88.12
C+	20.21	93.30	19.97	97.67

* Composite sample in the same bioreactor.

**Table 5 polymers-16-01908-t005:** Evaluation of compost maturity and plant growth.

Sample	RGeP ^a^ (%)	Biomass R ^b^	RGrP ^a^ (%)	Germination Index ^a^ (GI)
IS	100.0	134.8	105.5	105.5
EA	95.0	116.9	123.1	116.9
NL	102.6	88.3	100.9	103.5
AB	101.3	80.9	95.4	96.6
PTB *	63.9	90.0	84.9	54.2
PFB *	63.9	90.0	84.9	54.2
CAS	97.4	155.7	78.0	75.9
EB	81.0	115.5	83.5	67.6
EX	102.6	112.7	114.4	117.3
BP	102.6	151.5	125.9	129.1
C+	101.3	110.6	99.8	99.9

^a^: Germination index [40]. ^b^: Ecotoxicity test [18]. * Samples in the same bioreactor. RGeP: relative germination percentage; Biomass R: proportion of total plant biomass compared to the total biomass; RGrP: relative growth percentage.

## Data Availability

Data is contained within the article.

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
