# Peer review of "Assessment of the Biodegradability and Compostability of Finished Leathers: Analysis Using Spectroscopy and Thermal Methods"

_polymers, 2024, doi:10.3390/polym16131908_

Round 1

Reviewer 1 Report

Comments and Suggestions for Authors

The submitted article experimentally investigated the biodegradation properties of leather treated with different chemicals, polymers, resins, and different traditional finishing agents were evaluated using FTIR, thermal analysis (TGA) and solid state NMR spectroscopy methods. The results showed that polyurethane and acrylic wax coatings exhibited biodegradability, while traditional chemicals showed moderate levels of biodegradation. This study emphasizes the importance of selecting sustainable finishing chemicals to promote environmentally friendly leather production practices. The authors performed a good and interesting job concerning environmental remediation. The English throughout the Manuscript is good and the Manuscript is well structured. The following minor points can be addressed by the authors to improve their study:

- Use passive sentences instead of active sentences in scientific reports.

- Use the abbreviations in the rest of text instead of repeating the word associated with its abbreviation in parenthesis. The same for the text in Tables (use code instead of code and sample name simultaneously) as shown in Table 4.

- Use similar format for 1 without decimal and 1.0 with one decimal. In Table 3.

- The font of reference citation [34] and [35] within the text seems to be different.

- Use the better caption for Table 5.

- Line 280 to 313: Figure 1B was not written (indicated) in the text. Also add a vertical axis for this figure and increase the font size of horizontal axis. The same for the similar figures.

- It is suggested to read and cite the following good article somewhere relevant the promotion eco-friendly leather manufacturing in practices. https://doi.org/10.1007/s11356-021-18440-z.

Comments on the Quality of English Language

It is acceptable.

Reviewer 2 Report

Comments and Suggestions for Authors

The manuscript "Assessment of the Biodegradability and Compatibility of Finished Leathers: Analysis Using Spectroscopy and Thermal Methods" reports the behavior of different leather samples treated with different bio- and synthetic polymers under controllable conditions to evaluate the potential of biodegradation. The authors used spectroscopic (FT-IR and SC-NMR) and TGA analyses. They also performed composting and ecotoxicity experiments.

The overall composition of the paper is good, there is a correlation between the composition and the degradation process supported by experimental data. However, some microscopic technique (SEM) would have been very useful especially where the IR/ NMR indicated minor differences after biodegradation.

The following suggestions would be necessary to improve the present paper:

Title: the term "compatibility" is not reflected by the content of the work!

Keywords: replace "Fourier transform infrared spectroscopy; thermogravimetric analysis; solid-state nuclear magnetic resonance"- are not suggestive

Introduction:- delete the lines 69-82;

-lines 93-116 must be moved in Methods or Experimental

Section 2.3. delete the lines 165-167

Table 4. How was calculated the biodegradability (%)?

Table 5. F-ANOVA is significantly? Some details are need!

Section 3.3. - lines 286-292- the differences between spectra should be observed through subtract spectra function

- In all IR spectra the peaks are missing! Even the IR showed similar peaks there are some difference in the mass loss in TGA! So the biodegradation process should be studied by weighing the samples after exposure in the degradation conditions!

- there are differences in the intensity of the peaks! some cantitative studies would be useful!

-lines 335-337- these differences could be determined by changes in the crystallinity indexes calculated by IR  or XRD!

- Lines 347-349- the authors usually refer only to OH vibrations, the functional groups also need to be mentioned, there are some differences, even if only in intensity!

-lines 429-434- the phrases are similar in meaning!

-line 450-replace H bonds with H atoms!

-line 470- delete "This can be determined using the various techniques employed."

Based on these observations I recommend Major revision pf this paper!

Round 2

Reviewer 2 Report

Comments and Suggestions for Authors

The revised manuscript was improved, sufficient data have been provided and it is acceptable in the present form.